# Functional, Antioxidant, and Anti-Inflammatory Properties of Cricket Protein Concentrate (*Gryllus assimilis*)

**DOI:** 10.3390/biology11050776

**Published:** 2022-05-20

**Authors:** María Fernanda Quinteros, Jenny Martínez, Alejandra Barrionuevo, Marcelo Rojas, Wilman Carrillo

**Affiliations:** 1Posgrados, Universidad Técnica de Cotopaxi, Latacunga 050150, Ecuador; mquinteros@ueb.edu.ec; 2Departamento de Investigación, Universidad Estatal de Bolívar, Guaranda 020102, Ecuador; jmartinez@ueb.edu.ec (J.M.); abarrionuevo@ueb.edu.ec (A.B.); mrojas@ueb.edu.ec (M.R.); 3Departamento de Ingeniería Rural y Agroalimentaria, Universidad Politécnica de Valencia, 46022 Valencia, Spain

**Keywords:** *Gryllus assimilis*, cricket protein concentrate, antioxidant activity, functional properties, anti-inflammatory activity

## Abstract

**Simple Summary:**

The demand for consumption of protein of animal and vegetable origin increases with the growth of the world population. Therefore, new sources of animal protein that are nutritious, safe and sustainable for the environment are needed. In this situation, edible insects become a very attractive alternative due to their high protein content, their good functional and biological properties, and their environmental sustainability. Insects as food are not attractive to consumers who do not have them incorporated into their food culture, but products derived from them are accepted by consumers and can be used as functional ingredients by the food industry. Insect protein isolates and concentrates can be used as functional ingredients. The main objective of this work was to obtain protein concentrate from cricket (*Gryllus assimilis*) flour to evaluate their functional and biological properties. In addition to characterizing its protein profile and digestibility.

**Abstract:**

Edible insects can represent an alternative to obtain high-quality proteins with positive biological properties for human consumption. Cricket flour (*Gryllus assimilis*) was used to obtain cricket protein concentrate (CPC) using pHs (10.0 and 12.0) of extraction and pHs (3.0, 4.0, 5.0, and 6.0) of isoelectric precipitation (pI). Protein content, water and oil absorption capacity, protein solubility, antioxidant, and anti-inflammatory activities were determined. In addition, the protein profile was characterized by electrophoresis and the in vitro CPC digestibility was evaluated. Cricket flour presented 45.75% of protein content and CPC 12–5.0 presented a value of 71.16% protein content using the Dumas method. All samples were more soluble at pH 9.0 and 12.0. CPC 12–3.0 presented a percentage of water-binding capacity (WBC) of 41.25%. CPC 12–6.0 presented a percentage of oil-binding capacity (OBC) of 72.93%. All samples presented a high antioxidant and anti-inflammatory activity. CPC 12–4.0 presented a value FRAP of 70,034 umol trolox equivalents (TE)/g CPC, CPC 12–6.0 presented a value ABTS of 124,300 umol TE/g CPC and CPC 10–3.0 presented a DPPH value of 68,009 umol TE/g CPC. CPC 10–6.0 and CPC 12–6.0 presented high anti-inflammatory activity, with values of 93.55% and 93.15% of protection, respectively. CPCs can be used as functional ingredients in the food industry for their excellent functional and biological properties.

## 1. Introduction

It has been projected that by 2030 the world population could grow to around 8.6 billion and by the year 2050 it will grow to 9.8 billion people on the planet [1], with a demand of 70% of animal protein [2]. This situation suggests that different societies seek new sources of animal and vegetable protein with high nutritional quality that are also sustainable for the environment. Insects can be a good alternative for food protein with a sustainable production [3].

The consumption of insects in the human diet is called entomophagy and is practiced in many countries. There are about 2000 species of edible insects in the world. The type of insects consumed are worms, ants, wasps, bees, beetles, crickets, and grasshoppers [4], which are found in the diets of countries in Asia, Africa, and Latin America. In Australia, insects are also consumed in the diet of many regions [5,6,7,8]. Ants (*Atta laevigata*) are consumed in many places from Mexico to Argentina, and are known by different names, such as sikisapa (in Peru) and hormiga culona (in Colombia, Ecuador, and Argentina) [9]. This type of ant represents an important contribution of protein in the diet of Northwest Amazonian tribes, with 25% for the men’s diet and 32% for the women’s diet [10].

The consumption of insects in Europe is limited due to the rejection of consumers for this type of food. However, recently the European food industry has awakened its interest in this type of product since the Food and Agriculture Organization (FAO) published a report in 2015 promoting the consumption of insects as a source of protein for human and animal nutrition due to its nutritional value and its positive environmental aspects [11,12]. Moreover, the EFSA evaluated the risks of consuming insects. In addition, *Tenebrio molitor* larvae has been included in the list of insects that can be used as an ingredient in fish feed in the European Union (EU 2017/893) [13].

In the Ecuadorian Amazon, the indigenous communities have incorporated in their diet the consumption of larvae of *Rhynchophorus palmarum* L. (Coleoptera: Curculionidae), popularly called Chontacuro. At present, this consumption has become widespread in different areas of Ecuador because these insects are commercialized. These insects are seasonal and need palm trees (*Jesenia bataua*, *Maximiliana maripa*, and *Mauritia flexuosa*) for their life cycle [10]. In Latin America there is no specific legislation for the production and marketing of edible insects and their derivatives. Ecuador does not have specific legislation for the breeding and marketing of edible insects. Breeding farms comply with animal health standards and their products have to comply with Ecuador’s food safety and hygiene food standards. The company SARgrillos of Ecuador has been a pioneer in the breeding and commercialization of cricket flour (*Gryllus assimilis*). They have managed to introduce cricket flour and snacks in the Ecuadorian market. For this reason, we have decided to work with this species and look for new products that generate less consumer rejection, such as protein concentrates.

Edible insects have a high content of protein, lipids, fiber, and vitamins. Cricket flour with a protein content between 45–70% has been described in the literature, with 20.00–30% lipids and 5% fiber [14,15,16,17]. Insect meals and their protein fractions may be a way to introduce insects into human diets [18]. It is then important to know the solubility of the fractions, their functional properties, their biological properties, and their amino acid profile. Obtaining protein isolates and concentrates from cricket flour can facilitate their use as functional ingredients for insects in the food industry. Yi et al. (2013) [19] have described the functional properties of proteins from five insect species: *Tenebrio molitor*, *Zophobas morio*, *Alphitobius diaperinus*, *Acheta domesticus*, and *Blaptica dubia*. The extraction of proteins and fractions was done in an aqueous medium. A protein content between 50–75% was reported. Insect proteins were found to have the functional property of forming gels. This property may be important for their techno-functional applications in the food industry. Stone, Tanaka, and Nickerson (2019) [20] have described cricket and worm meal proteins with 65.50% and 66% protein content, respectively. Both protein extracts showed low solubility at pH 3.0, 5.0, and 7.0. These proteins extracts presented high digestibility and good water and oil absorption capacity. Ndiritu et al. (2019) [21] have described protein concentrate from *Acheta domesticus* with low protein solubility at pH 2.0–8.0. The functional properties were evaluated in the presence of NaCl, observing an increase in water absorption and a decrease in oil absorption.

The main objective of the work was to obtain cricket protein concentrate from cricket (*Gryllus assimilis*) flour by alkaline extraction followed by isoelectric precipitation and evaluation of its polyphenol content, functional, and biological properties.

## 2. Materials and Methods

### 2.1. Animal and Plant Materials

Freeze-dried crickets (*Gryllus assimilis*) were provided by the certified company SARgrillos de Quito-Ecuador. The crickets were ground in a Retsch brand mill, Cyclone Twister model. Once the flour was obtained, it was defatted with the help of hexane for 24 h. White quinoa flour (*Chenopodium quinoa* Willd) was also prepared.

### 2.2. Materials

Folin–Ciocalteu reagent, gallic acid standard, 2,20-Azino-bis-(3-ethylbenzothiazoline-6-sulfonicacid) (ABTS), 2,2-diphenyl-1-picrylhydrazyl (DPPH), 2,4,6-Tris(2-pyridyl)-s-triazine (TPTZ), and 6-Hydroxy-2,5,7,8-tetramethylchroman-2-carboxylic acid (Trolox standard) were obtained from Sigma-Aldrich (St. Louis, MO, USA). Analytical grade solvents and reagents were obtained from Sigma-Aldrich (St. Louis, MO, USA).

### 2.3. Proximal Analysis

The chemical composition of defatted cricket flour and quinoa flour were analyzed according to standardized protocols of the Association of Official Analytical Chemists (AOAC, 2012) [22]. Fat content was analyzed according to AOAC 2003.06:2012, moisture according to AOAC 925.10:2012, fiber content according at AOAC 962.09:2012, ashes according to AOAC 942.05:2012. The protein content was determined by the Dumas method using a conversion factor (Kp 5.60) [23] and the total carbohydrate content was calculated by difference.

### 2.4. Preparation of Cricket Protein Concentrate (CPC)

CPC was obtained following the methodology described by Vilcacundo et al. (2017) [24]. Cricket flour was prepared using 1000 freeze-dried crickets. The crickets were ground in a mill (Cyclone Mill Twister, Retsch, Haan, Germany) and 120 g of cricket flour was obtained. The cricket flour was defatted with hexane solvent in a ratio of 1:10 (*w*/*v*) under constant magnetic stirring for 3 h, the resulting suspension was vacuum filtered and air dried for 24 h. Then, 10 g of defatted cricket flour was dissolved in distilled water in a ratio of 1:10 (*w*/*v*). Then, the pH was adjusted to 10.0 and 12.0 with the help of 2N NaOH. The solution was centrifuged for 30 min at 6800× *g* at 5 °C using a centrifuge (Eppendorf 5804 R, Hamburg, Germany). The precipitate (fiber, sugars, starches, minerals) was discarded and the supernatant (soluble protein) was separated to adjust its pH to 3.0, 4.0, 5.0, and 6.0 with the help of 1N HCl. Then, the solution was centrifuged for 30 min at 6800× *g* at 5 °C to separate the precipitate. The pH of the precipitates was adjusted to 7.0 and it was frozen at −80 °C and then lyophilized using a lyophilizer (Christ Alpha 1–4 L Dplus, Germany) at −40°C to −60 °C for 4 days. The percentage of yield was determined gravimetrically as follows: (% Yield = CPC g/cricket flour g × 100).

### 2.5. Quantification of Protein Content of CPC

Protein content in CPCs was determined by the Dumas combustion method using an elemental analyzer (Vario Macro Cube, Elementar, Hanau, Germany). The equipment was calibrated with a sample of a sulphonamide standard. 20 mg of sample of lyophilized CPCs were placed in small aluminum capsules. The capsules were inserted into the sample injection mechanism to be transformed from nitrogen to its gaseous form by calcination. The assays were performed in triplicate and the protein percentage of the samples was calculated as: % Protein = F × %N, where F is the conversion factor 5.60 and %N is the percentage of nitrogen calculated by the equipment [23].

### 2.6. Functionals Properties of CPC

The functional properties of CPC that were evaluated were protein solubility, water absorption capacity, and oil absorption capacity. The methodology is described according to Pazmiño et al. (2018) [25].

#### 2.6.1. Protein Solubility

CPCs (0.2% *w*/*v*, based on weight protein content of each sample) were dissolved in distilled deionized water, and the pH of the suspension was adjusted to pH 3.0, 6.0, 9.0, and 12.0 using solutions 0.05 N HCl and NaOH. The suspensions were shaken for 1 h and centrifuged at 12,000× *g* for 10 min in a centrifuge (Eppendorf 5804 R, Hamburg, Germany). The content of protein in the supernatant was analyzed with the bicinchoninic acid (BCA) protein assay kit (Thermo Fisher Scientific, Waltham, MA, USA) using bovine serum albumin (BSA) as a standard protein at concentration of (0.125 to 2.0 mg/mL). Protein solubility was calculated using the following formula: % protein solubility = (protein content of supernatant/total protein content in the sample) × 100.

#### 2.6.2. Oil Absorption Capacity (OAC)

CPCs were dissolved in canola oil (1:10 ratio) in a pre-weighed tube. The suspensions were homogenized for 1 min using a vortex and then every 5 min until 30 min. Then, the suspensions were centrifuged at 2000× *g* for 15 min using a centrifuge (Eppendorf 5804 R, Hamburg, Germany). Then, the oil was drained, and the tube was tilted for 10 min and then weighed. OAC results were expressed as the content of oil absorbed per gram of sample.

#### 2.6.3. Water Absorption Capacity (WAC)

CPC were dissolved in distilled deionized water at 1:10 ratio in a pre-weighed tube. The mixture was homogenized for 30 s every 10 min for 5 times. Then, the mixture was centrifuged at 4000× *g* for 20 min using a centrifuge (Eppendorf 5804 R, Hamburg, Germany). The tubes were drained at 45° angle for 10 min and then weighed. WAC was calculated as the content of water absorbed by the weight of the protein sample.

### 2.7. Simulation of The Digestion Gastrointestinal in Vitro of CPCs

The CPC samples were subject to a gastric and duodenal simulated digestion process. 10 mg of CPCs were used to simulate the gastrointestinal digestion. The gastric phase was made at pH 3.0 for 120 min using porcine pepsin in simulated gastric fluid (0.035 M NaCl). The reaction was stopped by heating at 80 °C for 10 min. The duodenal phase was performed at pH 7.0 for 120 min. For this, 1 mL of CPC gastric hydrolysate was taken and mixed with 1 mL of pancreatin solution. The enzymatic reaction was stopped by heating at 80 °C for 10 min. The hydrolysates were frozen at −80 °C and lyophilized until use [26]. Three replicates of the digestion assays were performed [26].

### 2.8. Analysis by Electrophoresis of CPCs and Hydrolysates

CPCs and their gastrointestinal hydrolysates were characterized by sodium dodecyl sulphate polyacrylamide gel electrophoresis (SDS-PAGE). The samples were analyzed in a Miniprotean equipment (Bio-Rad, Hercules, CA, USA) at 200 V for 30 min using 12% polyacrylamide gels for CPC and 16% for hydrolysates. Molecular weights were calculated with the help of a molecular weight standard with a range of 2–250 kDa (Bio-Rad, CA, USA). Gels were stained with Coomassie blue G-250 solution for 24 h. Gels were photographed and processed on a gel analyzer (Analytik Jena Geltower, Thermo Fischer Scientific, Dublin, Ireland) [27].

### 2.9. Antioxidant Activity In Vitro of CPCs

#### 2.9.1. Quantification of Total Polyphenol Content (TPC)

CPCs and cricket flour samples were used to extract TPC for analysis. CPCs and cricket flour were dissolved in the mix of methanol: water (70:30 *v*:*v*). The samples were stirred for 5 min. Then, the sample was subjected to an ultrasonic bath using ColeParmer 8892-MTH (Cole-Parmer, Vernon Hills, IL, USA) and then centrifuged for 10 min. 1 mL of the solution was separated and mixed with 6 ml of distilled deionized water and 1 mL of Folin-Ciocalteau reagent. The mixture was left to rest for 3 min. Then, 2 mL of sodium carbonate (20%) was added and heated at 40 °C for 3 min. The absorbance of samples was measured at 765 nm on a Shimadzu model2600 spectrophotometer (Shimadzu, Kyoto, Japan). Standard calibration curve of gallic acid (GA) at 0–100 mg GA/L was used to determine content of TPC of the samples. The standard curve of GA obtained was (y = 0.0027x + 0.066, R^2^ = 0.997). TPC results obtained were expressed as mg gallic acid equivalents GAE/g of CPC, dry weight (DW) [28].

#### 2.9.2. Ferric-Reducing Antioxidant Power (FRAP) Method

The FRAP reagent was prepared by mixing (25 mL of 300 mM sodium acetate buffer at pH 3.60 + 2.50 mL of 10 mM TPTZ diluted in 40 mM HCl + 2.50 mL of 20 mM ferric chloride hexahydrate). 900 µL of FRAP reagent was mixed with 90 µL of distilled deionized water and 30 µL of CPC. The mixture was incubated in the dark at 37°C for 30 min. It was then centrifuged at 500 RPM for 5 min. Finally, the absorbance of the samples was measured at 593 nm. The trolox standard was used to make a calibration curve (100–500 umol) and the curve that was obtained was (y = 0.0017x − 0.149, R^2^ = 0.9976). The antioxidant activity data measured by FRAP were expressed as µmol trolox equivalents (TE)/g CPC, dry weight (DW) [29].

#### 2.9.3. 2-Azinobis (3-Ethyl-Benzothiazoline-6-Sulfonic Acid) Cation Bleaching ABTS Method

CPCs (200 μL) were mixed with 3800 μL of ABTS solution (7 mM ABTS solution + 2.45 mM potassium persulfate solution in a 1:1 ratio) and was incubated for 45 min in darkness. Afterwards, the mix was diluted adding phosphate buffer (pH 7.0) until obtaining an absorbance of 1.10 ± 0.01 at 743 nm. The Trolox standard solution (0 μmol to 500 μmol) was used as a standard curve to determine the concentrations of antioxidant. The standard curve obtained was (y = 0.014x + 0.2169, R^2^ = 0.9931). The results of antioxidant activity by ABTS assays were expressed as μmol of trolox equivalents (TE)/g CPC, dry weight (DW) [30].

#### 2.9.4. 2,2-Diphenyl-1-Picrylhydrazyl (DPPH) Radical Scavenging Assay

Antioxidant activity of CPCs was measured by the DPPH method described by Boeri et al. (2019). Trolox standard was used as the reference standard curve (0–800 µmol Trolox/L) and the calibration curve was obtained (y = 0.0007x, R^2^ = 0.9997). All assays were done three times (*n* = 3). The results obtained were represented as µmol trolox equivalents (TE)/g CPC, dry weight (DW) [31].

### 2.10. Anti-Inflammatory Activity In Vitro of CPC

The anti-inflammatory activity of CPC was evaluated using the membrane stabilization method described by Bouhlali et al. (2016) [32]. A sterile anticoagulant solution (0.05% citric acid, 0.42% sodium chloride, 0.80% sodium citrate, and 2% dextrose in distilled water) was prepared. 1 ml of anticoagulant solution was mixed with 1 mL of blood extracted from healthy volunteers who did not use anti-inflammatory drugs such as NSAIDs fifteen days after the extraction. The mixture was centrifuged at 6000× *g* for 30 min. The cell pellet was washed with isotonic saline solution (0.90%) and a 10% cell suspension was prepared. Subsequently, a reaction mixture containing 1 ml of phosphate buffer pH 7.0, 1 mL of CPC (20 mg/mL), 0.5 mL of blood cell suspension (10%), and 2 mL of hypotonic saline solution (0.36%) was prepared. The mixture was incubated for 30 min at 37 °C and centrifuged at 12,300× *g* for 10 min. The hemoglobin content of the supernatant was measured at 560 nm. The results were expressed as percentage of protection (%P) and calculated with the following formula: (% *p* = 100 − Absorbance sample/Absorbance control × 100).

### 2.11. Statistical Analysis

Results were presented in the article as mean ± standard deviation (SD). All trials had three replicates. Statistical differences of the samples used in this study were evaluated with one-way ANOVA analysis (*p* < 0.05) followed by the Tukey test. The statistical differences were expressed with different letters in the tables and figures. This statistical analysis was performed with the help of Stargraphics software.

## 3. Results and Discussion

Table 1 shows the results obtained from the proximal analysis of cricket flour. The cricket flour has a high protein content with 45.75%, followed by carbohydrates 20.80%, and 20% fat content. A great part of cricket flour is protein. Kiiru et al. (2020) [4] have described whole cricket flour with a protein content of 61.39% and defatted cricket flour with 68.48%. Calzada-Luna et al. (2021) have reported cricket flour (*Archeta domesticus*) with a percentage of 67.40% protein [33]. Adamková et al. (2017) have described a protein content in cricket flour (*Gryllus assimilis*) of 56% [34]. Laroche et al. (2019) have described a protein content of cricket flour (*Acheta domesticus*) with values of 58.30–78.50% and mealworm flour (*Tenebrio molitor*) with values between 48.70–75.40%. The flours were defatted by different methods [35].

The data obtained in this study is lower compared to the reported values in the literature for protein content in cricket flour. In this study, we have used the conversion factor (kp) of 5.60 recommended by Janssen et al. (2017) [23]. The conversion factor (Kp) of 6.25 is usually used to determine the protein content in edible insects and this can lead to an overestimation of the protein content. It should be taken into account that insects present non-protein nitrogen such as chitin, nucleic acids, phospholipids, and other products. Janssen et al. (2017) [23] have described *Tenebrio molitor* flour calculated with these two factors and differences were observed in the percentage of protein—ith a Kp of 6.25, 58.8% was obtained, and for a Kp of 5.60, a percentage of 44.8 was obtained.

The studies by Kiiru et al. (2020), Calzada-Luna et al. (2021), Adamková et al. (2017), and Laroche et al. (2019) have used a conversion factor of 6.25 to calculate the percentage of protein in the corresponding flours, for that reason they present higher values of protein content. The protein content of cricket flour was compared with the protein content of quinoa flour, as this is a widely consumed pseudocereal in Ecuador. Quinoa flour presented 13.63% protein, its content is very low when compared to cricket flour with its 45.70% protein, its value was 4.8 times higher than that of quinoa. Proximal composition of edible insects depends on its origin, stage of life, sex, and diet [23].

Table 2 shows % yield of CPCs obtained at different pHs. The CPCs treated at extraction pH 10.0 presented percentages between 19.07% and 36.18%, while the CPCs obtained at extraction pH 12.0 presented yield percentages between 11.56% and 72.75%. The sample 12–4.0 presented the highest percentage of yield. Statistical analysis shows no statistical differences between 12–3.0, 12–4.0, and 12–5.0. Similarly, no statistical differences were found between 10–3.0, 10–4.0, and 10–5.0 when compared between them. The extraction pH of 12.0 favors high yield percentages in the CPCs tested. Laroche et al. (2019) have described the yield percentage calculated for the protein extracts of *Acheta domesticus* and *Tenebrio molitor* with values (31.00%–38.90%) and (33.20%–39.30%) [35]. Nongonierma, Lamoureux, and FitzGerald (2018) have described obtaining cricket protein isolate (CPI) from (*Gryllus sigillatus*) with an alkaline extraction at pH 10.0 followed by adjustment to pH 7.0.%. Yield of CPC was 20.90% and the protein content was 57.04% [36]. The % yield of CPCs are consistent with those reported by other studies of edible insects. Yield percentages are affected by the pHs conditions used in alkaline extraction processes and the pHs used in isoelectric precipitation of proteins. The physicochemical characteristics of the proteins present in insects can be affected by these pHs.

Table 2 shows the protein content of CPC determined by the Dumas method. The protein content of the samples measured by the Dumas method presented values between 63.60% and 71.16%. The maximum value of 71.16% corresponds to the CPC 12–5.0 sample. The sample with the highest value of the CPC 10.0 group was the sample precipitated at pH 4.0 (CPC 10–4.0) while the sample with the highest value of the CPC 12.0 group was the sample precipitated at pH 5.0 (CPC 12–5.0). The CPC 12–6.0 sample was statistically different when compared to the others, while the rest of the samples did not present statistically significant differences between them. In general, the two groups of CPCs 10.0 and 12.0 presented protein content values greater than 60%.

Protein solubility and water and oil absorption capacity were the three functional properties of CPCs evaluated in this study. The effect of pH on protein solubility was analyzed (Figure 1). The protein solubility of the CPCs was determined at pH 3.0, 6.0, 9.0, and 12.0 by means of the BCA method, and the percentage of solubility was determined. Figure 1 shows the protein solubility percentages of the samples. In Figure 1A,B it can be seen that the CPCs extracted at pH 10.0 and pH 12.0 are more soluble at acidic and basic pH. They are less soluble at pH 6.0 at values close to neutral pH. At pH 9.0 and 12.0 it can be seen that the highest solubility percentage was achieved with percentages close to 50.00%. CPC 12–6.0 presented a value of 52.35% and 53.49%, respectively. However, at pH 6.0 the percentage of solubility of all samples dropped to a range from 6.05% to 16.88%, and all samples had values below 20%. This indicates that these CPCs are lower soluble at neutral pH. This property is important to know because this will allow us to know what possible uses these protein concentrates may have in the food industry. Statistical analysis showed that all treatments tested have significant statistical differences when compared to each other.

Ndiritu et al. (2019) have described that the greater solubility of proteins at high pHs (pH 12.0) may be due to the unfolding of the protein structure and the exposure of its hydrophilic groups. A high pHs improves the ionization of amino acids and produces an increase in protein solubility. They also describe that the decrease in protein solubility at pH 6.0 could be the result of reduced repulsion between amino acids, thus greater binding as the pH approaches the isoelectric point [21].

Protein solubility is affected by the extraction and precipitation pH used in the process of obtaining proteins. Insect proteins have their pI isoelectric point around pH 4.0. The pI of silkworm (*Bombyx mori*) and spider (*Nephila edulis*) proteins has been reported at pH values of 4.37–5.05 and 6.47, respectively [37]. In this study we have used isoelectric points between pH 3.0 and pH 6.0.

Low solubility of edible insect proteins at high pHs has been reported. Ndiritu et al. (2019) described protein concentrate from (*Acheta domesticus*) with high protein solubility at pH 12.0 (19.46%) [21]. Omotoso and Adedire, 2007, described high solubility of palm weevil proteins, (*Rhynchophorus phoenicis* f.) at pH 11.0 [38]. Omotoso, 2006, has described the solubility of the proteins of the larva of (*Cirina forda*) at pH 11.0 [39]. CPC can be used as analog of meat due to its solubility and can be used in the formulations of meat products like hamburgers.

The water absorption capacity (%WBC) of the CPCs was evaluated. Table 3 shows the %WBC values. Values in the ranges 23.40% to 41.25% were found, the highest value corresponding to the CPC 12–3.0 sample, and the lowest to the CPC 10–6.0 sample. Statistical analysis indicated that the CPC 10–6.0 and CPC 12–3.0 samples presented significant differences (*p* < 0.05) when compared to the other samples. This functional property is important when using CPCs as ingredients in the food industry.

The functional property called oil absorption capacity (%OBC) was also evaluated in the CPCs samples. Table 3 shows the OBC results of CPCs. The values are between the range of 27.10% to 72.93%. The lowest sample corresponds to the CPC 10–6.0 sample and the highest value corresponds to the CPC 12–6.0 sample. Statistical analysis indicates that all samples have significant differences (*p* < 0.05) when compared to each other. The samples of the CPC group extracted at pH 12.0 presented higher values comprised in the range of 56.67–72.93%, thus being the samples with the highest percentages of OBC. Bußler et al. (2016) [8] determined OBC and WBC of protein isolates of *Tenebrio molitor* and *Hermetia illucens*. They observed that the OBC functional property was higher for both isolates. Our OBC and WBC data have the same behavior, and the %OBC was higher than the %WBC.

CPCs and their gastrointestinal hydrolysates were characterized by the SDS-PAGE electrophoresis technique. Figure 2 shows the protein profile of the CPCs. The protein profile was established between the ranges of 37 KDa to 250 kDa. The following protein bands were found with these molecular weights 37 kDa, 45 kDa, 50 kDa, 75 KDa, 150 kDa, and 250 kDa. This profile with these intense bands was identical for all the CPCs analyzed. The band that corresponds to the molecular weight of 45 kDa was the one that showed the highest intensity of staining, which indicates that it is in higher concentration. CPC 12 samples showed low molecular weight bands of 5 kDa and 10 kDa.

Hall et al. (2017) described cricket protein extract (control) and cricket protein hydrolysate (CPH) with bands between molecular weights of 14.4 kDa and 212 kDa in cricket (*Gryllodes sigillatus*) [40]. CPH presented a band of molecular weights of 62.2 kDa and 45 kDa that were not appreciated in the control. The 45 kDa band was identified in the CPCs in this *G. assimilis* study. Hall et al., 2020, have described cricket protein extract (*Gryllodes sigillatus*) treated with microwaves [41]. They observed a protein profile between 6 kDa and 250 kDa for non-heat-treated and heat-treated cricket protein extract. Santiago et al. (2021) have described the protein profile of a protein isolate of *G. assimilis* with bands of 10 kDa, 15 kDa, 50 kDa, 75 kDa, 150 kDa, and 250 kDa [7]. The protein profile of the CPCs in this study is very similar to that reported by Santiago et al. (2021) and differentiates between the 37 kDa and 45 kDa bands, remembering that the band with the highest intensity was the 45 kDa band [7]. Bußler et al. (2016) [8] have characterized the protein profile of meal worm (*Tenebrio molitor*) and black soldier fly (*Hermetia illucens*) with bands between 13 kDa and 250 kDa. Different studies have characterized insect proteins. The bands observed between the 14 kDa and 32 kDa ranges have been described as cuticle proteins or 24 kDa chymotrypsin-like proteinase. The bands between the 32 kDa and 90 kDa ranges can give rise to the bands (43 kDa) melanization-inhibiting protein, (59 kDa) described as β-glycosidase and (59 kDa) and the band of (85 kDa) melanization-inhibiting protein, engaging protein. Bands greater than 95 KDa have been described as vitellogening-like protein [42,43,44,45,46,47,48].

Gastrointestinal hydrolysates of CPCs were also characterized by SDS-PAGE electrophoresis (Figure 3). Figure 3 compares the profile of unhydrolyzed CPC and the gastrointestinal hydrolysates of CPCs. It can be seen that the profile of CPC proteins changes completely when subject to a gastric or duodenal hydrolysis process. In both stages of hydrolysis, the proteins were degraded. The gel analysis software does not identify bands in the hydrolysates. This indicates that cricket proteins are highly susceptible to being hydrolyzed by pepsin and by the mixture of pepsin and pancreatin under the conditions tested in this study. Hall et al. (2020) have described two hydrolysates of cricket (*Gryllodes sigillatus*) obtained with the enzyme alcalase [41]. A conventional hydrolysate with the enzyme alcalase and another hydrolysate pre-heated with microwaves and then hydrolyzed with the same enzyme. Both hydrolysates were analyzed by electrophoresis and no bands were observed in the gel compared to the control with bands between 6 kDa and 250 kDa. Hydrolysis of CPCS was complete, as in our study.

Table 4 shows the content of TPC determined in the CPCs and the cricket flour. The cricket flour presented a TPC value of 942 mg GAE/100 g sample, while the CPCs presented values with a range of 557 mg GAE/100 g sample and 1546 mg GAE/100 g sample, the lowest value corresponds to the CPC sample 10–6.0 and the highest value corresponds to the CPC sample 12–6.0. In general, the TPC values are quite high, but the CPC samples 10–3.0, CPC 10–4.0, CPC 10–5.0, and CPC 12–5.0 presented lower values than cricket flour. Statistical analysis indicates that samples CPC 10–3.0, CPC 10–4.0, CPC 12–3.0, CPC 12–4.0, and cricket flour did not present statistical differences when compared to each other. Kim et al. (2020) have described cricket TPC (*Gryllus bimaculatus*). The cricket flour presented a TPC value of 661 mg GAE/100 g sample and the protein concentrate obtained presented a value of 1218 mg GAE/100 g sample [49]. In our study, we reported higher values of TPC content described by the aforementioned group. Musundire et al. (2014 a) have described the TPC content in edible bug (*Encosternum delegorguei*) extracts [50]. They described the content of TPC in insects thermally treated with cooking with a value of 3600 mg GAE/100 g sample and 2800 mg GAE/100 g sample for insects not thermally treated. Both values were higher than those reported in the present study for *G. assimilis*. Musundire et al. (2014b) [51] have described the TPC content of crickets from Zimbabwe (*Henicus whellani*) with a TPC value of 7700 mg GAE/100 g sample, this value was also higher than our TPC values. The authors indicate that crickets (*Henicus whellani*) are able to absorb phenolic compounds from plants, and then are able to sequester or metabolize them. Del Hierro et al. (2020) [52] have described the TPC content in extracts obtained from house cricket (*Acheta domesticus*) and mealworm (*Tenebrio molitor*) with values of 5000 mg GAE/100 g sample and 3800 mg GAE/100 g sample, respectively. Both extracts were obtained with a mixture of ethanol and water. These extracts described the above-presented TPC values as higher than the CPCs values of this study. These differences in polyphenol content may be due to the difference in insect species described in the different studies and the differences in their diets on insect farms.

The antioxidant activity of CPCs was evaluated using three different methods such as the FRAP, ABTS, and DPPH methods. Table 4 shows the antioxidant values of the CPCs evaluated with the FRAP method. It was observed that all the samples presented a high antioxidant activity with values ranging from 28,062 µmol TE/g CPC to 70,034 µmol TE/g CPC. The sample with the highest value was CPC 10–3.0 and the sample with the lowest value was CPC 12–5.0. The analysis of variance obtained through the Stargraphics statistical package showed that the combination of the solubilization pH vs. the precipitation pH are factors that influence the antioxidant activity of the CPCs, showing significant differences between the treatments. Like the antioxidant activity values measured by the FRAP method, the antioxidant activity values of the CPCs measured with the ABTS method were very high values. The results of the antioxidant activity of CPC measured by the ABTS method showed values between 58,059 µmol TE/g CPC and 124,300 µmol TE/g CPC. The sample with the highest value was CPC 12–6.0 and the sample with the lowest value was CPC 10–6.0. The statistical analysis reflected that there are statistical differences when compared between them. CPC 12–3.0 and CPC 12–4.0 were not statistically different when the two were compared (Table 4).

The third method used to evaluate the antioxidant activity of CPCs was the DPPH method. Table 4 shows the antioxidant activity values obtained with this method. It can be seen that all the samples tested showed high antioxidant activity with the DPPH method. The values obtained are in the range of 21,699 µmol TE/g CPC to 68,009 µmol TE/g CPC. The highest value corresponds to the CPC 10–3.0 sample and the lowest value corresponds to the CPC 10–6.0 sample. Both samples belong to the group of isolates extracted at pH 10.0.

De Matos et al. (2021) [53] have described antioxidant activity of a protein concentrate called BCPC (*G. assimilis*) and of hydrolysates of this concentrate obtained with the enzymes flavourzyme, alcalase, and neutrase. BCPC presented a FRAP value of 357 µmol TE/g sample and the hydrolysate with the highest activity presented a value of 688 µmol TE/g sample. ABTS presented values of the values obtained in this study were much higher than those previously mentioned. These differences in activity could be due to differences between the protein concentrates. De Matos et al. (2021) [53] obtained BCPC by alkaline extraction at pH 11.0 followed by isoelectric precipitation at pH 4.5. In addition, the samples could have different concentrations of phenolic compounds, but in this study, they were not determined.

The anti-inflammatory activity of the CPC samples was evaluated with an in vitro method called membrane stabilization. Figure 4 shows the protection percentages of the CPCs tested. The CPC 10–3.0 and CPC 12–3.0 samples presented the lowest values with protection percentages of 0.40% P and 14.52% P. The CPC 10–6.0 and CPC 12–6.0 samples presented the highest protection values of 93.52% P and 93.15% P. As in the previous case, the two studied variables converge: pH of extraction and pH of isoelectric precipitation. CPC 10–5.0 (92.34% P) and CPC 10–6.0 (93.52% P; CPC 12–6.0 (93.15% P) presented higher values than the positive control tested, which was diclofenac, an anti-inflammatory type of fast-acting NSAIDs, which presented a percentage of protection of 87.90% P. Nino et al. (2021) [54] have described the role of phenolic compounds identified in numerous edible insects and their role in biological activities such as antioxidant and anti-inflammatory activity. They indicate that these phenolic compounds are introduced into the insect organism through its herbivore diet. Through a process called sclerotization, insects include phenolic compounds in their cuticle with the help of proteins and enzymes. There are few works where the TPC content has been determined and the molecules identified in edible insects. This assumption, that the compounds phenolics are introduced through the animal’s diet, has been discussed by different authors [55,56,57,58,59]. For example, Ferreres et al. (2008) [60] described different phenolic compounds, such as phenolic acids, ferulic, sinapic, and p-coumaric, of extracts of large white butterfly (*Pieris brassicae*) reared on turnips (*Brassica rapa* var. rapa L). The high antioxidant and anti-inflammatory activity of CPC (*G. assimilis*) reported in this study may be due to the presence of various phenolic compounds assimilated through the plant diet. Protein isolates and concentrates are widely used in the food and pharmaceutical industry for their techno-functional properties. Textured soy products (TPS) are widely used as analogous meat substitutes because they contain a high percentage of protein (70–90%) and can be stored for long periods of time at room temperature and only need hydration for their use as food in meats, soups, and other foods [61,62]. Ecuador is not a high producer of soybeans and therefore needs to look for alternatives to reduce the importation of TPS. The CPCs could be a good alternative to replace this type of product.

## 4. Conclusions

In the near future, insects may become an important source of protein for the human diet and their production is also environmentally sustainable. Many consumers have a certain reluctance to include insects in their diet, therefore their derivatives are an excellent alternative so that they can be used by the food industry. These transformations can be in the form of protein concentrates. In addition to containing a high protein content and high gastrointestinal digestibility, CPCs have added biological values such as a strong antioxidant and anti-inflammatory power which could be due to their high content of phenolic compounds. The functional properties allow evaluating the possible uses of the isolates as ingredients in food products. In future research, the different proteins of the CPC, the peptides of the hydrolysates, and the phenolic compounds could be identified with the help of mass spectrometry.

## Figures and Tables

**Figure 1 biology-11-00776-f001:**
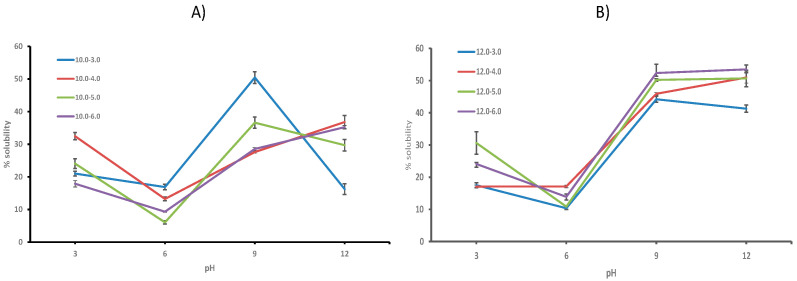
Protein solubility of cricket protein (CPC) at pHs 3.0, 6.0, 9.0, and 12.0.

**Figure 2 biology-11-00776-f002:**
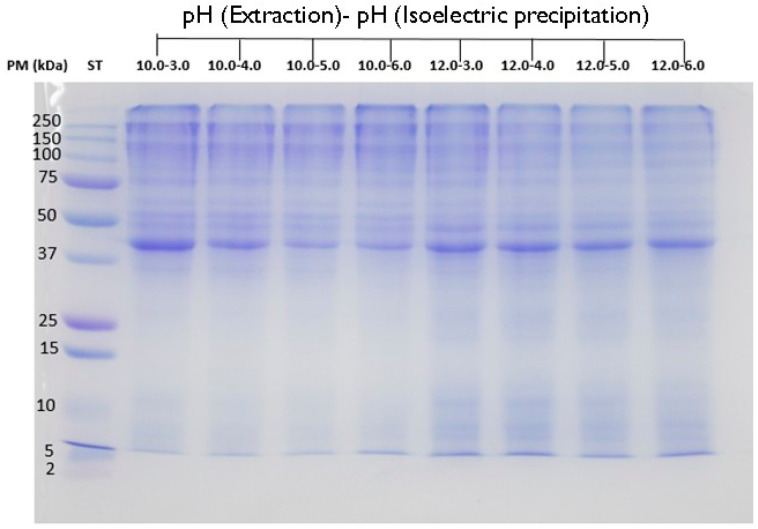
Profile of proteins of cricket protein concentrate (CPC) by SDS-PAGE. The uncropped western blot figures were presented in Appendix A.

**Figure 3 biology-11-00776-f003:**
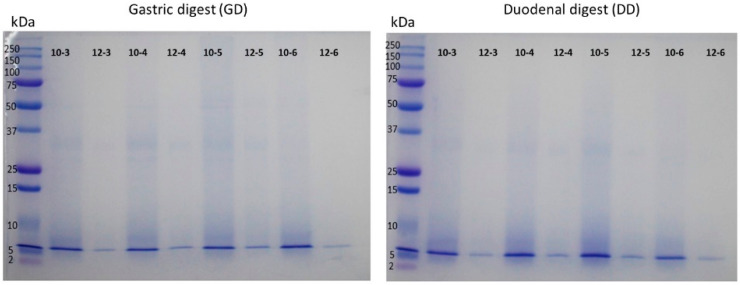
Analysis by SDP-PAGE of gastrointestinal digest of cricket protein concentrate (CPC). The uncropped western blot figures were presented in Appendix A.

**Figure 4 biology-11-00776-f004:**
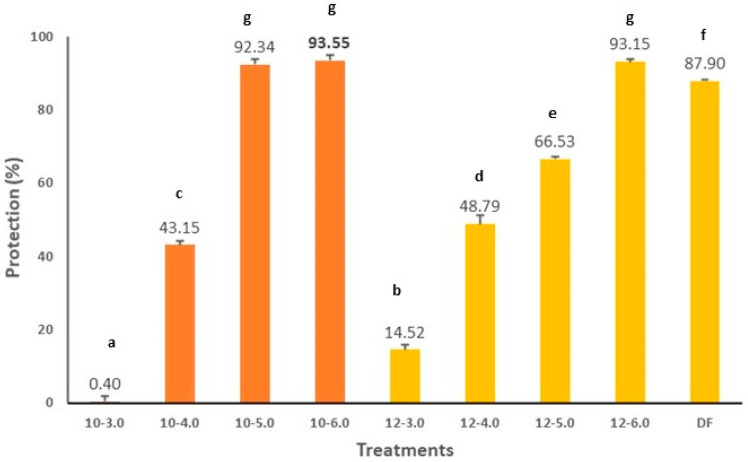
Anti-inflammatory activity of cricket protein concentrate (CPC). Results were expressed as mean ± standard deviation (*n* = 3) and were evaluated by one-way Anova and Turkey test (*p* < 0.05). Statistical differences were indicated with different letters. DF (diclofenac) control.

**Table 1 biology-11-00776-t001:** Proximal composition of flour (*Gryllus assimilis*) and quinoa flour. Calculated by ratio g/100 g dry matter.

Assay	Cricket	Quinoa
Protein	45.75 ± 2.25	13.63 ± 1.75
Fat	20.00 ± 1.94	4.20 ± 0.22
Fiber	5.01 ± 0.07	0.76 ± 0.81
Ashes	4.94 ± 0.18	2.08 ± 0.14
Moisture	3.50 ± 0.05	10.02 ± 1.30
Carbohydrates	20.80 ± 0.06	69.31 ± 0.07

Results were expressed as mean ± standard deviation (*n* = 3).

**Table 2 biology-11-00776-t002:** % Yield of cricket protein concentrate (CPC) obtained at different pH of extraction and pHs of isoelectric precipitation and % protein of CPC by Dumas method.

CPC	% Yield for 20 g Sample	% Protein
10.0–3.0	33.92 ± 1.53 ^b^	66.86 ± 0.68 ^b,c^
10.0–4.0	36.18 ± 2.68 ^b^	69.37 ± 0.92 ^d,e,f^
10.0–5.0	32.28 ± 6.05 ^b^	69.10 ± 0.59 ^d,e^
10.0–6.0	19.07 ± 4.90 ^a^	65.74 ± 0.20 ^b,c^
12.0–3.0	67.05 ± 3.32 ^c^	68.06 ± 0.17 ^c,d^
12.0–4.0	72.75 ± 6.18 ^c^	70.80 ± 1.19 ^e,f^
12.0–5.0	67.77 ± 2.39 ^c^	71.16 ± 0.51 ^f^
12.0–6.0	11.56 ± 4.90 ^a^	63.60 ± 0.40 ^a^

Results were expressed as mean ± standard deviation (*n* = 3) and were evaluated by one-way Anova and Turkey test (*p* < 0.05). Statistical differences were indicated with different superscripts letters.

**Table 3 biology-11-00776-t003:** % Water-binding capacity (WBC) and oil-binding capacity (OBC) of CPCs.

CPC	% Water-Binding Capacity (WBC)	% Oil-Binding Capacity (OBC)
10.0–3.0	29.76 ± 0.49 ^b^	67.55 ± 2.28 ^d^
10.0–4.0	25.11 ± 0.95 ^a,b^	51.70 ± 1.62 ^b^
10.0–5.0	25.29 ± 3.23 ^a,b^	51.90 ± 1.72 ^b^
10.0–6.0	23.40 ± 1.87 ^a^	27.10 ± 0.09 ^a^
12.0–3.0	41.25 ± 0.98 ^c^	59.03 ± 1.83 ^c^
12.0–4.0	28.92 ± 1.58 ^a,b^	56.67 ± 2.57 ^b,c^
12.0–5.0	28.18 ± 3.73 ^a,b^	59.36 ± 2.04 ^c^
12.0–6.0	27.66 ± 0.28 ^a,b^	72.93 ± 2.04 ^d^
Cricket flour	29.76 ± 0.49 ^b^	67.55 ± 2.28 ^d^

Results were expressed as mean ± standard deviation (*n* = 3) and were evaluated by one-way Anova and Turkey test (*p* < 0.05). Statistical differences were indicated with different superscripts letters.

**Table 4 biology-11-00776-t004:** Antioxidant activity of cricket protein concentrate (CPC) using FRAP, ABTS, and DPPH methods.

CPC.	TPC mg GAE/100 g Sample	FRAP umol TE/g CPC	ABTS umol TE/g CPC	DPPH umol TE/g CPC
10.0–3.0	972 ± 21.54 ^c^	65154 ± 863 ^d^	100336 ± 3665 ^f^	68009 ± 1979 ^c^
10.0–4.0	909 ± 43.00 ^c^	70034 ± 2586 ^e^	85533 ± 1015 ^e^	63913 ± 2615 ^c^
10.0–5.0	606 ± 21.54 ^b^	39825 ± 431 ^c^	61351 ± 508 ^b^	27446 ± 5942 ^a^
10.0–6.0	557 ± 0.00 ^a^	39776 ± 1790 ^c^	58059 ± 0.00 ^a^	21699 ± 7112 ^a^
12.0–3.0	924 ± 21.48 ^c^	31252 ± 745 ^a,b^	68487 ± 507 ^c^	22233 ± 1710 ^a^
12.0–4.0	969 ± 56.81 ^c^	34219 ± 745 ^b^	69038 ± 506 ^c^	31908 ± 3558 ^a^
12.0–5.0	940 ± 37.26 ^c^	28062 ± 431 ^a^	63605 ± 1759 ^b^	31401 ± 2616 ^a^
12.0–6.0	1546 ± 38.49 ^d^	35063 ± 1543 ^b^	124300 ± 1817 ^g^	46992 ± 1769 ^b^
Cricket flour	942 ± 37.33 ^c^	35603 ± 1230 ^b^	34897 ± 881 ^h^	69020 ± 990 ^a^

Results were expressed as mean ± standard deviation (*n* = 3) and were evaluated by one-way Anova and Turkey test (*p* < 0.05). Statistical differences were indicated with different superscripts letters. TPC (total polyphenol content), GAE (gallic acid equivalents), and TE (trolox equivalents).

## Data Availability

Not applicable.

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
