# Peer review of "Functional, Antioxidant, and Anti-Inflammatory Properties of Cricket Protein Concentrate (*Gryllus assimilis*)"

_biology, 2022, doi:10.3390/biology11050776_

Round 1

Reviewer 1 Report

The paper presented by the authors is entitled "Cricket protein isolate (Gryllus assimilis) as a new functional ingredient with antioxidant and anti-inflammatory activities.

The work presented is interesting. It evaluates the techno-functional properties of an insect that can find an application in the food industry for people who do not consume whole insects. It also evaluates the biological properties (anti-oxidant and anti-inflammatory) that can be applied in the medical field to people suffering from specific ailments. It is an original article, which could, however, be improved. Some comments are given in the following lines.

TITLE

The title of the article can be reduced and modified. As the maximum protein content of the different extracts is less than 90%, the author should refer to "protein concentrate" instead of "protein isolate".

The proposed title is therefore: "Functional, antioxidant, and anti-inflammatory properties of cricket protein concentrate (Gryllus assimilis)".

SUMMARY

The author should add in one sentence the conclusion of his work in the abstract.

INTRODUCTION

There are several species of insects consumed in Ecuador. The author must justify why he/she chose to work specifically on Gryllus assimilis. In addition, the author should provide information on the nutritional composition of this insect, citing the authors of the works. The author also chooses to talk about European legislation on insects, even though they were collected in Ecuador. Is there no legislation in Latin America?

MATERIAL AND METHODS

The methodology section lacks many details.

Line 86-89: How many insects were collected? What volume of hexane was used for delipidation?

Line 103-112: The author should specify the equipment used for centrifugation. The author should give the centrifugation speed in "g" and not "RPM". Give the brand of freeze-dryer used and the freeze-drying conditions.

According to the work of Janssen et al. (2017), the conversion factor that should be used for the calculation of the protein content of insect extracts is 5.60. The author uses a conversion factor of 5.60 for the calculation of the protein content of insect extracts. However, the author uses a factor of 6.25, which overestimates the amount of protein. Shouldn't it simply be changed? Read the article: Janssen, R.H., Vincken, J.P., Van den Broek, L.A.M., Fogliano, V. and Lakemond, C.M.M. Nitrogen-to-Protein Conversion Factors for Three Edible Insects: Tenebrio molitor, Alphitobius diaperinus, and Hermetia illucens. Journal of Agricultural and Food Chemistry. 2017, 65, 2275-2278. DOI: 10.1021/acs.jafc.7b00471

Line 122-125: The author should give the detailed methodology.

Line 144-154: The author should specify the quantities of samples, reagents, and volume of solvent used.

Lines 155–178: Explain what FRAP, ABTS, and DPPH mean. It would have been interesting to have a control for these different tests in order to better appreciate the properties that will be observed. Perhaps vitamin C.

RESULTS AND DISCUSSION

Line 202: Delete "As you can see".

Lines 200–215: Nutrient content should be expressed as g/100 g of fresh matter or g/100 g of dry matter, not as a percentage. Also to be amended in Table 1.

The protein content of the isolates is less than 80%, which means that this isolate could contain other components such as lipids, carbohydrates, or minerals. It would have been interesting to have an idea of these other constituents (proximal composition), which could certainly influence the functional properties of the isolates.

Whether at pH 10 or pH 12, the protein extraction yields and precipitated protein content are the lowest. How does the author explain this?

Neither in the methodology nor in the results is it clearly stated how the calculation of protein solubility was performed. The authors should clarify this. This solubility should be calculated in relation to the total protein content.

Figure 1: The resolution of the figure is very low and needs to be improved. The solubility calculation should have been done at successive pH's (3, 4, 5, 6, 7, 8, 9, 10, 11, 12) to get a fairly real isoelectric pH value.

Some of the results presented are just compared with those in the literature, but are not discussed or explained. The authors should correct this. 

Table 4: The author should be able to compare the antioxidant properties of protein isolates with those of a known antioxidant.

Author Response

Reviewer # 1

Comments and Suggestions for Authors

The paper presented by the authors is entitled "Cricket protein isolate (Gryllus assimilis) as a new functional ingredient with antioxidant and anti-inflammatory activities.

The work presented is interesting. It evaluates the techno-functional properties of an insect that can find an application in the food industry for people who do not consume whole insects. It also evaluates the biological properties (anti-oxidant and anti-inflammatory) that can be applied in the medical field to people suffering from specific ailments. It is an original article, which could, however, be improved. Some comments are given in the following lines.

 TITLE

The title of the article can be reduced and modified. As the maximum protein content of the different extracts is less than 90%, the author should refer to "protein concentrate" instead of "protein isolate".

The proposed title is therefore: "Functional, antioxidant, and anti-inflammatory properties of cricket protein concentrate (Gryllus assimilis)".

Answer: the title was changed for "Functional, antioxidant, and anti-inflammatory properties of cricket protein concentrate (Gryllus assimilis)".

Cricket protein isolate (CPI) was changed for cricket protein concentrate (CPC) in the manuscript, tables and figures.

SUMMARY

The author should add in one sentence the conclusion of his work in the abstract.

Answer: this sentence was added “CPCs can be used as functional ingredients in the food industry for their good functional and biological properties”.

INTRODUCTION

There are several species of insects consumed in Ecuador. The author must justify why he/she chose to work specifically on Gryllus assimilis. In addition, the author should provide information on the nutritional composition of this insect, citing the authors of the works. The author also chooses to talk about European legislation on insects, even though they were collected in Ecuador. Is there no legislation in Latin America?

In the Ecuadorian Amazon, the indigenous communities have incorporated in their diet the consumption of larvae of Rhynchophorus palmarum L. (Coleoptera: Curculionidae) popularly called Chontacuro. At present, this consumption has become widespread in different areas of Ecuador because these insects are commercialized. These insects are seasonal and need palm trees (Jesenia bataua, Maximiliana maripa and Mauritia flexuosa) for their life cycle [10]. In Latin America there is no specific legislation for the production and marketing of edible insects and their derivatives. Ecuador does not have specific legislation for the breeding and marketing of edible insects. Breeding farms comply with animal health standards and their products have to comply with Ecuador's food safety and hygiene food standards. The company SARgrillos of Ecuador has been a pioneer in the breeding and commercialization of cricket flour (Gryllus assimilis). They have managed to introduce cricket flour and snacks in the Ecuadorian market. For this reason, we have decided to work with this species and look for new products that generate less consumer rejection, such as protein concentrates.      

MATERIAL AND METHODS

The methodology section lacks many details.

Line 86-89: How many insects were collected? What volume of hexane was used for delipidation?

Answer: this paragraph was changed for: CPC was obtained following the methodology described by Vilcacundo et al. (2017) [24]. Cricket flour was prepared using 1,000 freeze-dried crickets. The crickets were ground in a mill (Cyclone Mill Twister, Retsch, Haan, Germany) and 120 g of cricket flour were obtained. The cricket flour was defatted with hexane solvent in a ratio of 1:10 (w/v) under constant magnetic stirring for 3 h, the resulting suspension was vacuum filtered and air dried for 24 h. Then, 10g of defatted cricket flour was dissolved in distilled water in a ratio of 1:10 (w/v). Then the pH was adjusted to 10.0 and 12.0 with the help of 2N NaOH. The solution was centrifuged for 30 min at 6800 x g at 5°C using a centrifuge (Eppendorf 5804 R, Hamburg, Germany). The precipitate (fiber, sugars, starches, minerals) was discarded and the supernatant (soluble protein) was separated to adjust its pH to 3.0; 4.0; 5.0 and 6.0 with the help of 1N HCl. Then, the solution was centrifuged for 30 min at 6800 x g at 5°C to separate the precipitate. The pH of the precipitates was adjusted to 7.0 and it was frozen at -80°C and then lyophilized using a lyophilizer (Christ Alpha 1-4 L Dplus, Germany) at -40°C to -60°C for 4 days. The percentage of yield was determined gravimetrically as follows: [% Yield = CPC g/ cricket flour g x 100].

Line 103-112: The author should specify the equipment used for centrifugation. The author should give the centrifugation speed in "g" and not "RPM". Give the brand of freeze-dryer used and the freeze-drying conditions.

Answer: this paragraph was modified for “CPC was obtained following the methodology described by Vilcacundo et al. (2017) [24]. Cricket flour was prepared using 1,000 freeze-dried crickets. The crickets were ground in a mill (Cyclone Mill Twister, Retsch, Haan, Germany) and 120 g of cricket flour were obtained. The cricket flour was defatted with hexane solvent in a ratio of 1:10 (w/v) under constant magnetic stirring for 3 h, the resulting suspension was vacuum filtered and air dried for 24 h. Then, 10g of defatted cricket flour was dissolved in distilled water in a ratio of 1:10 (w/v). Then the pH was adjusted to 10.0 and 12.0 with the help of 2N NaOH. The solution was centrifuged for 30 min at 6800 x g at 5°C using a centrifuge (Eppendorf 5804 R, Hamburg, Germany). The precipitate (fiber, sugars, starches, minerals) was discarded and the supernatant (soluble protein) was separated to adjust its pH to 3.0; 4.0; 5.0 and 6.0 with the help of 1N HCl. Then, the solution was centrifuged for 30 min at 6800 x g at 5°C to separate the precipitate. The pH of the precipitates was adjusted to 7.0 and it was frozen at -80°C and then lyophilized using a lyophilizer (Christ Alpha 1-4 L Dplus, Germany) at -40°C to -60°C for 4 days. The percentage of yield was determined gravimetrically as follows: [% Yield = CPC g/ cricket flour g x 100]”.

According to the work of Janssen et al. (2017), the conversion factor that should be used for the calculation of the protein content of insect extracts is 5.60. The author uses a conversion factor of 5.60 for the calculation of the protein content of insect extracts. However, the author uses a factor of 6.25, which overestimates the amount of protein. Shouldn't it simply be changed? Read the article: Janssen, R.H., Vincken, J.P., Van den Broek, L.A.M., Fogliano, V. and Lakemond, C.M.M. Nitrogen-to-Protein Conversion Factors for Three Edible Insects: Tenebrio molitor, Alphitobius diaperinus, and Hermetia illucens. Journal of Agricultural and Food Chemistry. 2017, 65, 2275-2278. DOI: 10.1021/acs.jafc.7b00471.

Answer: Thank you very much for indicating this important factor for the work. We change the conversion factor from 6.25 to 5.60 and change the values in Table 2, with the change of their standard deviations. These changes affected the abstract, results and discussion and the bibliographical references. In the results we have discussed the difference between the values found in the literature (Kp 6.25) and our protein values with Kp 5.60. In the bibliography we have included the quote from Janssen et al. (2017).

Line 122-125: The author should give the detailed methodology.

Answer: these methodologies were detailed

Functionals properties of CPC

The functional properties of CPC that were evaluated were protein solubility, water absorption capacity, oil absorption capacity. The methodology is described according at Pazmiño et al. (2018) [25].

2.6.1. Protein solubility

CPCs (0.2% w/v, based on weight protein content of each sample) were dissolved in distilled deionized water, and the pH of the suspension was adjusted to pH 3.0; 6.0; 9.0 and 12.0 using solutions 0.05 N HCl and NaOH. The suspensions were shaken for 1 h and centrifuged at 12,000 x g for 10 min in a centrifuge (Eppendorf 5804 R, Hamburg, Germany). The content of protein in the supernatant was analyzed with the bicinchoninic acid (BCA) protein assay kit (Thermo Fisher Scientific, Germany) using bovine serum albumin (BSA) as a standard protein at concentration of (0.125 to 2.0 mg/ml). Protein solubility was calculated using the following formula: % protein solubility = (protein content of supernatant/total protein content in the sample) × 100.

2.62. Oil absorption capacity (OAC)

CPCs were dissolved in canola oil (1:10 ratio) in a pre-weighed tube. The suspensions were homogenized for 1 min using a vortex and then every 5 min until 30 min. Then, the suspensions were centrifuged at 2,000 x g for 15 min using a centrifuge (Eppendorf 5804 R, Hamburg, Germany). Then, the oil was drained, and the tube was tilted for 10 min and then weighed. OAC results were expressed as the content of oil absorbed per gram of sample.

2.63. Water absorption capacity (WAC)

CPC were dissolved in distilled deionized water at 1:10 ratio in a pre-weighed tube. The mixture was homogenized for 30 s every 10 min for 5 times. Then, the mixture was centrifuged at 4,000 x g for 20 min using a centrifuge (Eppendorf 5804 R, Hamburg, Germany). The tubes were drained at 45° angle for 10 min and then weighed. WAC was calculated as the content of water absorbed by the weight of the protein sample

Line 144-154: The author should specify the quantities of samples, reagents, and volume of solvent used.

Answer: This section was modified

“CPCs and cricket flour samples were used to extract TPC for analysis. CPCs and cricket flour were dissolved in the mix of methanol: water (70:30 v: v). After, the samples were stirred for 5 min. Then, the sample was subjected to an ultrasonic bath using ColeParmer 8892-MTH (Cole-Parmer, Illinois, USA) and then centrifuged for 10 min. 1 ml of the solution was separated and mixed with 6 ml of distilled water and 1 ml of Folin-Ciocalteau reagent. The mixture was left to rest for 3 min. After, 2 ml of sodium carbonate (20%) was added and heated at 40°C for 3 min. The absorbance of samples was measured at 765 nm on a Shimadzu model2600 spectrophotometer (Shimadzu, Kyoto, Japan). Standard calibration curve of gallic acid (GA) at 0–100 mg GA/L was used to determine content of TPC of the samples. The standard curve of GA obtained was (y = 0.0027x + 0.066, R2=0.997). TPC results obtained were expressed as mg gallic acid equivalents GAE/g of CPC, dry weight (DW) [28]”.

Lines 155–178: Explain what FRAP, ABTS, and DPPH mean. It would have been interesting to have a control for these different tests in order to better appreciate the properties that will be observed. Perhaps vitamin C.

Answer: FRAP, ABTS and FRAP were defined in the titles. We do not include control for other antioxidants because standard Trolox was used for the calibration curves.

RESULTS AND DISCUSSION

Line 202: Delete "As you can see".

Answer: This sentence was deleted

Lines 200–215: Nutrient content should be expressed as g/100 g of fresh matter or g/100 g of dry matter, not as a percentage. Also to be amended in Table 1.

Answer: The legend was changed for “Table 1. Proximal composition of flour (Gryllus assimilis) and quinoa flour. Calculated by ratio g/100 g dry matter”

The protein content of the isolates is less than 80%, which means that this isolate could contain other components such as lipids, carbohydrates, or minerals. It would have been interesting to have an idea of these other constituents (proximal composition), which could certainly influence the functional properties of the isolates.

Answer: Other tests would have to be carried out to know the type of carbohydrates that the concentrates contain. Phenolic compounds influence the biological activities of CPCs. Other functional activities of CPCs would have to be evaluated to determine the possible influence of carbohydrates and fats. The profile of fatty acids could also be analyzed to determine the content of saturated and unsaturated fatty acids and to know their influence on the conservation of the samples because fatty acids can be oxidized. We plan to carry out these tests in other future studies.

Whether at pH 10 or pH 12, the protein extraction yields and precipitated protein content are the lowest. How does the author explain this?

Answer: Yield percentages are average figures when compared to other authors. Protein percentages are between 60-70%, being normal values for protein concentrates.

Neither in the methodology nor in the results is it clearly stated how the calculation of protein solubility was performed. The authors should clarify this. This solubility should be calculated in relation to the total protein content.

Answer: In the methodology section, we further clarified in detail how the three functional properties tested were calculated.

Figure 1: The resolution of the figure is very low and needs to be improved. The solubility calculation should have been done at successive pH's (3, 4, 5, 6, 7, 8, 9, 10, 11, 12) to get a fairly real isoelectric pH value.

Answer: Figure 1 was changed and that error has been corrected. The pHs for calculating solubility were 3, 6, 9 and 12. In addition, error bars were added to the figure.

Some of the results presented are just compared with those in the literature, but are not discussed or explained. The authors should correct this.

Answer: Some aspects of the job have been discussed more like protein content.

Table 4: The author should be able to compare the antioxidant properties of protein isolates with those of a known antioxidant.

Answer: Other antioxidant molecules were not included in the antioxidant activity assays. Only the standard trolox was used for the calibration curve. Trolox is a potent antioxidant used for this type of assay and activity is calculated based on its activity.

Reviewer 2 Report

In this manuscript entitled "Cricket protein isolate (Gryllus assimilis) as a new functional ingredient with antioxidant and anti-inflammatory activities", the authors evaluate the nutritional composition, functional properties, and digestibility of cricket protein isolate in vitro study because of the background of the need for new animal protein sources that are also environmentally friendly. I think the data presented is valuable. I have comments, explained below. I hope that my comments are very useful for the improvement of this research.

Major comments
(1)    Novelty of the study: I don't see what is novel about this study. Would the crickets you are using be different from previous studies? The discussion often expresses that the same results were obtained as in the previous data. I could not understand the novelty even after reading through the paper. Therefore, please clearly indicate the novelty of this study in the Introduction section.
(2)    Value: The number of digits in a value is too large. For example, the value shown on line 53 is at most two digits. Also, I think the value (70034.49 umol trolox equivalents (TE)/g) shown in line 31 is about 3 significant digits in different units. Other values also have too large digits. Please correct them to the appropriate number of digits throughout the MS.
(3)    CPI: Why was the CPI prepared for this study and not cricket flour? The reason should be indicated. In addition, please indicate why authors decided to examine the items evaluated in this study.
(4)    Digestibility: In this study, protein digestibility is evaluated by electrophoresis images. However, evaluating digestibility from electrophoresis images is qualitative. Since digestibility is a quantitative concept, it would be better to measure the total amino acid content of the artificial digestive solution.
(5)    Discussion: In this study, authors were evaluating the properties of cricket proteins extracted at various pH. However, there is no consideration of which extraction method is best for digestibility, function, and nutritional property. It would be nice to have such a consideration

Minor comments
(6)    L97: If it was defatted, please indicate the method of defatting.
(7)    Figure 1: This should be shown in a bar graph. Also, please do some statistical processing.
(8)    L270-272: It needs to be explained in more detail why it can be used in meat products if it is soluble in alkali pH.
(9)    Fig.3: Is the amount of each protein being subjected to electrophoresis the same? I do not appear to have the same amount of protein in each lane.

Author Response

Reviewer #2

In this manuscript entitled "Cricket protein isolate (Gryllus assimilis) as a new functional ingredient with antioxidant and anti-inflammatory activities", the authors evaluate the nutritional composition, functional properties, and digestibility of cricket protein isolate in vitro study because of the background of the need for new animal protein sources that are also environmentally friendly. I think the data presented is valuable. I have comments, explained below. I hope that my comments are very useful for the improvement of this research.

 Major comments

Novelty of the study: I don't see what is novel about this study. Would the crickets you are using be different from previous studies? The discussion often expresses that the same results were obtained as in the previous data. I could not understand the novelty even after reading through the paper. Therefore, please clearly indicate the novelty of this study in the Introduction section.

Answer: Most studies on edible insects are based on characterizing the functional and biological properties of flours. The isolates are obtained to be hydrolyzed and to evaluate the properties of the hydrolysates. Our study is mainly based on characterizing the functional properties of the isolates that were obtained at pH 10 and 12 and pI 3.0-6.0. We have used a wider range of pHs than those commonly used in other species of edible insects. There are few studies on protein isolates of Gryllus assimilis. There are very few studies on the anti-inflammatory activity of protein isolates from crickets. The existing articles evaluated the biological properties of the flours and their hydrolysates.

Value: The number of digits in a value is too large. For example, the value shown on line 53 is at most two digits. Also, I think the value (70034.49 umol trolox equivalents (TE)/g) shown in line 31 is about 3 significant digits in different units. Other values also have too large digits. Please correct them to the appropriate number of digits throughout the MS.

Answer: Thanks for the comment. This error was corrected throughout the manuscript. Decimals for large figures are not necessary because the standard deviations are high. We have left two decimal places for the small numbers considering the standard deviation.

CPI: Why was the CPI prepared for this study and not cricket flour? The reason should be indicated. In addition, please indicate why authors decided to examine the items evaluated in this study.

Answer: The SARgrillos company in Ecuador markets the flour and a series of snacks based on the flour. The company wants to have new products on the market and thought about the possibility of producing cricket protein isolates and concentrates. Most of the studies in the bibliography are made on cricket flour. Protein isolates are made as possible functional ingredients for snacks, cookies and their direct sale as isolates for food use.

.The following has been included in the article: “Protein isolates and concentrates are widely used in the food and pharmaceutical industry for their techno-functional properties. Textured soy products (TPS) are widely used as analogous meat substitutes because they contain a high percentage of protein (50 to over 90%) and can be stored for long periods of time at room temperature and only need hydration for their use as food in meats, soups and other foods [61,62]. Ecuador is not a high producer of soybeans and therefore needs to look for alternatives to reduce the importation of TPS. The CPCs could be a good alternative to replace this type of product”. Two new references were added.

  1. Singh, P.; Kumar, R.; Sabapathy, S. N.; Bawa, A. S. Functional and edible uses of soy protein products. Compr. Rev. Food Sci. Food Saf. 2008, 7, 14-28.
  2. Jooyandeh, H. Soy products as healthy and functional foods. Middle East J. Sci. Res. 2011, 7, 71-80.

Digestibility: In this study, protein digestibility is evaluated by electrophoresis images. However, evaluating digestibility from electrophoresis images is qualitative. Since digestibility is a quantitative concept, it would be better to measure the total amino acid content of the artificial digestive solution.

Answer: We qualitatively analyzed protein digestion (presence and absence of bands stained with Coomassie blue). The gel documenter tells us whether or not there are bands on the gel within the range of 2 kDa-250 kDa. 16% polyacrylamide gels were used so that the movement of the bands is slow, the pore size is smaller and thus avoid losses due to size.

(5)    Discussion: In this study, authors were evaluating the properties of cricket proteins extracted at various pH. However, there is no consideration of which extraction method is best for digestibility, function, and nutritional property. It would be nice to have such a consideration

Answer: In the discussion it was expanded that high pHs favor protein solubility and the highest values of WAC and OAC were observed at pH 12. Protein digestibility and antioxidant activity are not affected by the extraction pH method.

Minor comments

L97: If it was defatted, please indicate the method of defatting.

Answer: this was added “CPC was obtained following the methodology described by Vilcacundo et al. (2017) [24]. Cricket flour was prepared using 1,000 freeze-dried crickets. The crickets were ground in a mill (Cyclone Mill Twister, Retsch, Haan, Germany) and 120 g of cricket flour were obtained. The cricket flour was defatted with hexane solvent in a ratio of 1:10 (w/v) under constant magnetic stirring for 3 h, the resulting suspension was vacuum filtered and air dried for 24 h. Then, 10g of defatted cricket flour was dissolved in distilled water in a ratio of 1:10 (w/v). Then the pH was adjusted to 10.0 and 12.0 with the help of 2N NaOH. The solution was centrifuged for 30 min at 6800 x g at 5°C using a centrifuge (Eppendorf 5804 R, Hamburg, Germany). The precipitate (fiber, sugars, starches, minerals) was discarded and the supernatant (soluble protein) was separated to adjust its pH to 3.0; 4.0; 5.0 and 6.0 with the help of 1N HCl. Then, the solution was centrifuged for 30 min at 6800 x g at 5°C to separate the precipitate. The pH of the precipitates was adjusted to 7.0 and it was frozen at -80°C and then lyophilized using a lyophilizer (Christ Alpha 1-4 L Dplus, Germany) at -40°C to -60°C for 4 days. The percentage of yield was determined gravimetrically as follows: [% Yield = CPC g/ cricket flour g x 100]”.

(7)    Figure 1: This should be shown in a bar graph. Also, please do some statistical processing.

Answer: The figure 1 was modified. The error bars were added in the figure.

(8)    L270-272: It needs to be explained in more detail why it can be used in meat products if it is soluble in alkali pH.

Answer: this sentence was changed for “CPI can be used as analog of meat due to its solubility and can be used in the formulations of meat products like hamburgers”.

(9)    Fig.3: Is the amount of each protein being subjected to electrophoresis the same? I do not appear to have the same amount of protein in each lane.

Answer: The analyzer of gels instrument does not appreciate significant differences. It does not recognize bands in either of the two gels. A trace is observed in the gels that may be due to interference from other molecules such as polyphenols, salts and minerals. The description was made based on the report of the instrument.

Round 2

Reviewer 1 Report

The responses to the various comments, as well as the various changes to the article, are adequate.

Reviewer 2 Report

I am satisfied with the revisions that have been made by the authors.